# NMR Diffusiometry Spectroscopy, a Novel Technique for Monitoring the Micro-Modifications in Bitumen Ageing

**Paolino Caputo [1]** , **Dlshad Shaikhah [2,3]** , **Michele Porto [1,*]** , **Valeria Loise [1,*]** , **Maria Penelope De Santo [4] and Cesare Oliviero Rossi [1]**

[1]  Department of Chemistry and Chemical Technologies, University of Calabria, 87036 Arcavacata di Rende (CS), Italy; paolino.caputo@unical.it (P.C.); cesare.oliviero@unical.it (C.O.R.)

[2]  Institute of Functional Surfaces, School of Mechanical Engineering, University of Leeds, Woodhouse Lane LS2 9JT, UK; D.M.Shaikhah@leeds.ac.uk

[3]  Department of Chemistry, College of Science, Salahaddin University-Erbil, Erbil 44002, Kurdistan

[4]  Department of Physics and CNR-Nanotec, University of Calabria, via Bucci 31C, 87036 Arcavacata di Rende (CS), Italy; maria.desanto@fis.unical.it

*   Correspondence: michele.porto@unical.it (M.P.); valeria.loise@unical.it (V.L.); Tel./Fax: +39-0984492045

**Abstract:** In the past three decades, several conventional methods have been employed for characterizing the bitumen ageing phenomenon, such as rheological testing, ultraviolet testing, gel permeation chromatography (GPC), gas chromatography (GC), atomic force microscopy (AFM), X-ray scattering, and Fourier transform infrared spectroscopy (FTIR). Nevertheless, these techniques can provide only limited observations of the structural micro-modifications occurring during bitumen ageing. In this study, Fourier transform nuclear magnetic resonance self-diffusion coefficient (FT-NMR-SDC) spectroscopy, as a novel method, was employed to investigate and compare the microstructural changes between virgin bitumen (pristine bitumen) and aged bitumen. The virgin bitumen was aged artificially using two standard ageing tests: Rolling Thin-Film Oven Test (RTFOT) and Pressure Ageing Vessel (PAV). For a comprehensive comparison and an assessment of the validity of this method, the generated samples were studied using various methods: rheological test, atomic force microscopy, and optical microscopy. Significant differences were obtained between the structure and ageing patterns of virgin and aged bitumen. The results indicate that the modification of maltenes to asphaltenes is responsible for the ageing character. When compared with the other methods' findings, FT-NMR-SDC observations confirm that the asphaltene content increases during ageing processes.

**Keywords:** aged bitumen; NMR diffusiometry; rheology; AFM

## 1. Introduction

The most widely used road material in the world is asphalt. As this product possesses the desired industrial characteristics (waterproof and excellent thermoplasticity), it is widely employed in the construction industry, mainly in road construction and other paved areas [1]. From the chemical point of view, an asphalt is defined as a heterogenous system consisting of macro-meter-sized inorganic particles, known as aggregates, and a binder material called bitumen [2]. Bitumen is a heavy hydrocarbon material, and it is the by-product of the fractural refinement process of crude oil, which removes the lighter fractions (i.e., liquid petroleum gas, gasoline, and diesel) [3].

Traditionally, bitumen is defined as a colloidal system consisting of micelles of high polarity and molecular mass known as asphaltene; these are the solid particles behaving as adhesive aggregates.

Richardson defined the asphaltenes as insoluble in naphtha but soluble in carbon tetrachloride and also introduced the term "carbenes" for the fraction insoluble in carbon tetrachloride but soluble in carbon disulphide [4]. The word "carboids" for the part insoluble in carbon disulphide is also seldom used [5] (although not used by Richardson [4]). In all cases, these two additional fractions are present in very limited amounts in paving-grade bitumen [6] and are generally not mentioned in the road industry. Asphaltene aggregates remain in an oily apolar environment of lower molecular weight, known as maltenes, granting fluidity [7]. The apolar maltene phase, in turn, is composed of saturated paraffins, aromatic oils, and resins. Figure 1 shows a schematic splitting of bitumen into its main components. The proposed model of bitumen in the literature is that the asphaltene, in the form of polar nano-aggregates, remains dispersed as a peninsula or group of peninsulas within a more apolar continuous maltene phase [8]. In the conventional colloidal model, the resins-to-asphaltenes ratio characterizes the bitumen behavior to a specific path if the bitumen is in solution (sol). Sol bitumen is due to a high maltene concentration that leads to rapid relaxation modes (a liquid-like system), while the bitumen is gelatinous (gel) when the asphaltene concentration is dominant, resulting in slow relaxation modes [9].

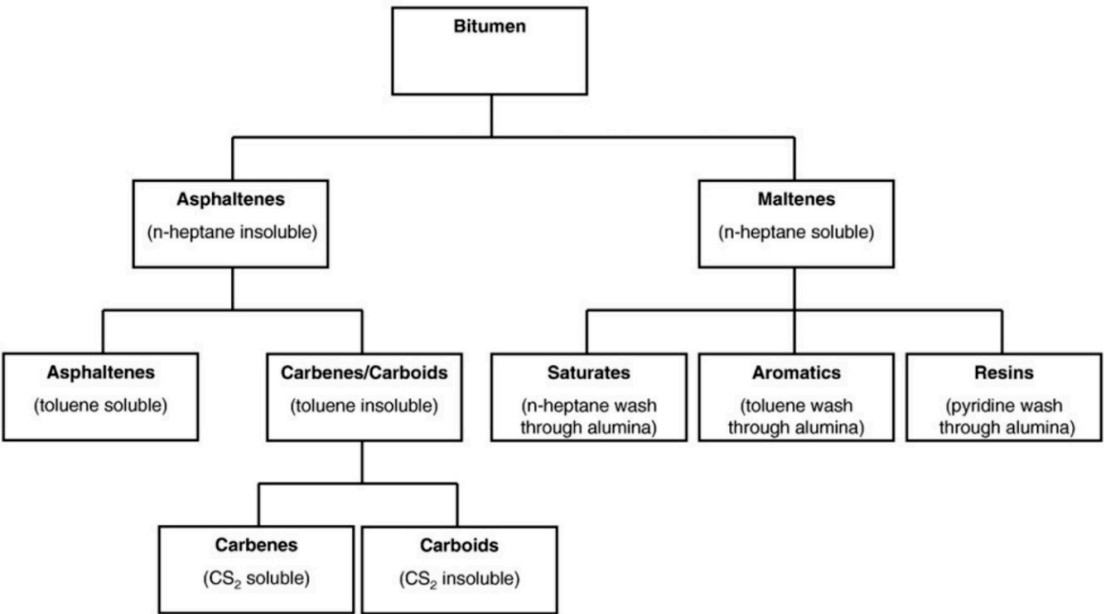

**Figure 1.** Bitumen separation into its various constituents, highlighting the SARA (saturates, aromatics, resins, asphaltenes) distribution, reproduced from [10], Copyright Elsevier, 2020.

In bitumen, resins are the dispersing agents for asphaltene molecules, combining them with aromatics and saturates; this produces the conditions for asphaltic flow. In fact, the chemistry of bitumen is the key element to defining its physical properties: following the analogy with reversed micelles in water-in-oil microemulsions [11], the stabilization of the polar domains is of pivotal importance for determining both the structure and the properties of the overall aggregates of organic-based materials [12,13], even if, it must be pointed out, the stabilization mechanism is quite general, spanning from organic materials to inorganic complexes [14] and even nanoparticles [15]. Due to these strict relationships between intermolecular interactions, the aggregates' structure, and their dynamic properties, the rheology (ductility at a given temperature/frequency) and behavior of bitumen are dependent not only on its structure, but also on the maltene's glass transition temperature and the effective asphaltene content [10].

When it comes to the bitumen structure, the microstructural model offers a typical point of view. It describes the system as a complex solution of evenly distributed molecules, classifying them according to their molecular weight and polarity; these behaviors are the key to grasping whether

the bitumen is modified. As there are millions of such constituents within the bitumen, its chemical analysis is usually performed based on the molecular structure type, not by studying the constituents individually [16]. Any technique, therefore, that differentiates the types of molecules or fractions within narrower properties would be a more effective form of analysis.

The stiffness and viscosity of asphalt pavement and its corresponding bitumen increase with time while it is in service; this is called the ageing phenomenon. As this phenomenon is responsible for the chemical modifications occurring in each fraction of bitumen, it is one of the most important factors impacting the bitumen structure. Asphalt research is currently directed towards the construction of new road pavements using milled reclaimed asphalt pavement (RAP) with consequent advantages both economically and environmentally. To use RAP in a bituminous conglomerate, it is necessary to add additives termed "Rejuvenating". The search for these additives is simply based on the evaluation of their effect on the mechanical performance of the final conglomerate. Having more information on the effect of the bitumen ageing process will make it possible to design these additives by identifying molecules containing functional groups which can interact in a targeted manner with aged bitumen components, thus regenerating them.

Generally, the ageing process is divided into two stages: short-term ageing and long-term ageing [17]. The former results from the loss of volatile components from the bitumen's maltene due to mixing at high temperature in the pristine paving process or during asphalt construction. The latter, in turn, occurs in the field, owing to the following factors: oxidation of bitumen components by atmospheric oxygen, evaporation of low-molecular-weight components of maltene, and polymerization between the bitumen's components. These processes induce greater stiffness and viscosity of the bitumen, which ultimately hardens the material, causing potential cracking and loss of its binding efficiency [16].

In the laboratory, different artificial standard methods for simulating the ageing phenomenon have been introduced. The Rolling Thin-Film Oven Test (RTFOT) and Thin-Film Oven Test (TFOT) techniques are considered short-term ageing as they simulate bitumen ageing during storage, mixing, transport, and placing as pavement. Meanwhile, to simulate long-term ageing, ultraviolet testing (UV) and Pressure Ageing Vessel (PAV) are employed. In this study, RTFOT and PAV are used as short- and long-term ageing simulation techniques, respectively [18].

As the outcome of ageing is crucial, a vast number of studies, using various perspectives and theories, have explored a range of parameters or factors relating to this phenomenon. Different in situ techniques have been used in the laboratory to evaluate, analyze, and identify what actually occurs inside bitumen during the ageing process [9]. These studies or techniques, however, are mainly measuring a specific factor of a material; this raises a challenge to identify all the feasible variables and forms of the material to study the bitumen ageing process, for either short-term ageing or long-term ageing.

The ageing phenomenon impacts not only the rheology of binders but also the structural chemistry; thus, there are numerous techniques to characterize this phenomenon, such as softening point (EN 1427 [19]), penetration test (EN 1426 [20]), and viscosity (EN 13302 [21]). As explained above, the process of ageing causes an increase in stiffness and viscosity but reduces the bitumen penetration grade. Thus, from this point of view, it is feasible to establish an ageing index by measuring the variation of the abovementioned physical properties (before and after ageing). This parameter has proven to be the most exemplary measure with regard to the results observed in the field [22].

It is important to note that assessing bitumen's physical and mechanical properties allows indirect macro-structural analysis of the binder. To measure these properties, the most commonly used techniques are dynamic shear rheology (DSR), softening point test, penetration test, and Brookfield or rotational viscosity determination [9]. Additionally, a variety of perspectives are utilized through a range of evaluation techniques to explain changes in asphalt materials after ageing. These techniques include Fourier transform infrared spectroscopy (FTIR) [23], spectrophotometry [24], X-ray scattering spectroscopy [25,26], gel permeation chromatography (GPC) [27], thin-layer chromatography with

flame ionization detection (TLC-FID) [28], atomic force spectroscopy (AFM) [29] and self-diffusion Pulsed Field Gradient Spin-Echo (PGSE) nuclear magnetic resonance (NMR) [30]. NMR spectroscopy is one of the most authentic and efficient techniques used to characterize complex materials i.e., bitumen. It is a conventional tool in the characterization of synthetic and natural products, wherein the structural and conformational behavior of their flexible molecules are investigated using anisotropic media [31]. In dynamic NMR characterization, the determination of self-diffusion coefficients can provide data about self-assembly [32,33], molecular dynamics, and spatial dimensions of cavities [34] and aggregates [35–38].

In this study, Fourier transform (FT) NMR self-diffusion coefficient measurements were performed on bitumen samples in order to have a better understanding of their molecular mobility and microstructure modification due to ageing; then, the measurements were compared with the results of rheological testing, AFM, and optical microscopy for validation. Ultimately, obtaining more information on the ageing process will provide consequent advantages in the design of rejuvenating additives and in understanding the structural organization of aged bitumen.

## 2. Materials

The pristine bitumen (PB) was kindly supplied by Total Italia S.p.A. (Italy) and produced in Saudi Arabia. It was used as a fresh standard with penetration grade 50/70 as measured by the usual standardized procedure [39]. Essentially, the standard needle is loaded with a weight of 100 g, and the length travelled into the bitumen specimen is measured in tenths of a millimeter for a known time, at a fixed temperature.

The chemical composition of the bitumen in terms of saturates, aromatics, resins, and asphaltenes (SARA) is reported in Table 1.

**Table 1.** Group composition of pristine bitumen.

| Sample | % *w/w* |
|:---:|:---:|
| Saturates | 4 |
| Aromatics | 51 |
| Resins | 22 |
| Asphaltenes | 23 |

### 2.1. Sample Preparation

In order to produce simulated short-term and long-term ageing samples, pristine bitumen (PB) was aged artificially via two conventional techniques: Rolling Thin-Film Oven Test (RTFOT) and Pressure Ageing Vessel (PAV). The RTFOT simulation was used for three ageing terms (75 min, 150 min, and 225 min) as per ASTM D2872, whereas the PAV simulation was performed on the virgin binder for a long term as specified in ASTM D6521. The sample IDs of RTFOT-aged binders were adopted based on their ageing time (BRTFOT-75, BRTFOT-150, and BRTFOT-225), while the PAV-aged binder was named BPAV.

### 2.2. Asphaltene and Maltene Separation (Modified Conventional Method)

To perform optical microscopy measurements, asphaltenes were isolated from the bitumen in the manner described elsewhere [40]. Chloroform and n-pentane were used as solvents to separate the maltenes and the asphaltenes.

## 3. Methodology

### 3.1. Rheological Characterization

The performance grades of the binders were determined using a dynamic shear rheometer (DSR) to quantify the viscoelastic properties in the temperature range 25–100 °C. The rheological

measurements were performed by utilizing a shear-stress-controlled rheometer (DSR5000, Rheometrics Scientific, Piscataway, USA), set up with a plate geometry (gap = 2 mm and diameter $\emptyset$ = 25 mm). The temperature of the system was controlled by a Peltier system ($\pm 1$ °C).

Temperature sweep (time cure) rheological tests were performed to analyze the mechanical response of modified bitumens versus temperature. Bitumen exhibits aspects of both elastic and viscous behaviors and is thus classified as a viscoelastic material [41,42]. DSR is a common and standard technique used to study the rheology of asphalt binders at high and intermediate temperatures [43,44].

Operatively, a bitumen sample was sandwiched between two parallel plates, one standing and one oscillatory. The oscillating plate was rotated accordingly with the sample and the resulting shear stress was measured.

During the tests, a periodic sinusoidal stress at constant frequency of 1 Hz was applied to the sample, and the resulting sinusoidal strain was measured in terms of amplitude and phase angle as the loss tangent (tan $\delta$).

All experiments were conducted during the heating of the bitumen sample. The study of rheological characteristics, complex shear modulus, and phase angle was performed in a time cure test (1 Hz) with a heating ramp rate of 1 °C/min. Information on the linear viscoelastic character of materials was provided by small-amplitude dynamic tests through characterization of the complex shear modulus [45]:

$$G^*(\omega) = G'(\omega) + iG''(\omega) \tag{1}$$

where $G^*(\omega)$ is the complex shear modulus, $G'(\omega)$ is the storage (in-phase) component (Pascal, Pa), $G''(\omega)$ is the loss (out-of-phase) component (Pascal, Pa), and $i$ is the imaginary parameter of the complex number. The definitions of the parameters are as follows:

- $G'(\omega)$ represents elastic and reversible energy;
- $G''(\omega)$ represents irreversible viscous dissipation of mechanical energy.

The linear response regime was acquired by reducing the applied stress amplitude for the viscoelastic measurements. All rheological analyses were achieved by applying stress within the viscoelastic region.

### 3.2. Optical Microscopy

A polarized Leica Digital Microscope Light Polarized (DMLP) Research Microscope equipped with a Leica DFC280 digital camera was utilized to track the melting behaviors and morphologies of the asphaltenes. A Mettler FP82 HThot stage with a Mettler FP90 temperature controller was applied to control the temperature profile. The sample was kept at constant temperature, for each investigated temperature, for 10 min before taking images.

### 3.3. Nuclear Magnetic Resonance (NMR) Characterization

NMR characterization was conducted using a Bruker 300 spectrometer. A range of temperatures—from 90 to 130 °C in 10 °C increments—was chosen to conduct NMR experiments to determine the self-diffusion coefficient (D) for each binder.

Using Fourier transform, the acquired NMR spectra were derived from the free induction decay (FID). In Figure 2, a typical $^1$H-NMR spectrum of the bitumen is presented. The number of scans used in the pulsed NMR experiment was 8 with pulse width equal to $\pi/2$. The measurements of D were conducted using a Diff30 NMR probe. As the transverse relaxation time ($T_2$) is much shorter than the longitudinal relaxation time ($T_1$), the Pulsed Gradient Stimulated Spin Echo (PFG-STE) sequence was utilized [46,47]. This sequence comprises three radiofrequency (RF) pulses of 90 ($\pi/2$-$\tau_1$-$\pi/2$-$\tau_m$-$\pi/2$) and two gradient pulses that are performed after the first and third pulse RF. $\tau_1$ and $\tau_m$ are the time

intervals between RF pulses (milliseconds). The echo appears at $2\tau_1 + \tau_m$, and the ECHO amplitude attenuation was derived from the equation below:

$$I\left(2\tau_1 + \tau_m\right) = I_0 e^{-\left[\frac{\tau_m}{T_1} + \frac{3\tau_1}{T_2} + (\gamma g \delta)^2 \, D(\Delta - \frac{\delta}{3})\right]} \tag{2}$$

where D is the self-diffusion coefficient, and $I$ and $I_0$ are the signal intensities in the presence and absence of gradient pulses. The NMR characterization parameters applied in the experiments to investigate the samples were as follows: $\delta$ (2 ms), the gradient length pulse; $\Delta$, diffusion delay time (30 ms); and $g$, the gradient amplitude (from 100 to 900 gauss/cm). The number of scans increased due to an increment in the number of repetitions. This NMR has a very low fitting standard deviation and good reproducibility of measurements, where the uncertainty of D is approximately 3%.

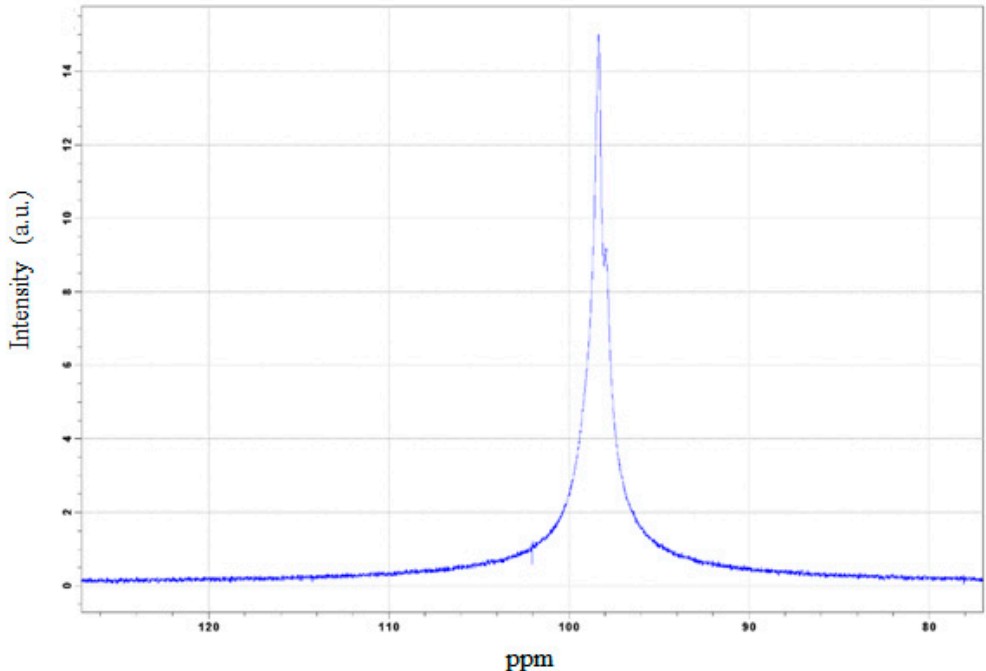

**Figure 2.** FT-NMR spin echo spectrum for the virgin bitumen.

According to the colloidal model, bitumen is generally composed of two principal constituents: asphaltene and maltene. The asphaltene is the rigid and polar part, which is characterized by high melting points. The maltene, in turn, is the soft and oily part that disperses the asphaltenes. Considering the low $T_2$ relaxation times of the asphaltenes [48], the self-diffusion coefficients can be attributed to the oily part of the bitumen; in fact, the NMR signal of asphaltenes relaxes during the pulse's application [49].

### 3.4. Atomic Force Microscopy (AFM)

Atomic force microscopy (AFM) was carried out using a Nanoscope VIII Bruker microscope which was operated in tapping mode. In this mode, the cantilever oscillates close to its resonance frequency (150 kHz) [50]. Since the cantilever oscillates up and down, the tip is in contact with the sample surface intermittently. When the tip is brought close to the surface, the vibration of the cantilever is influenced by the tip–sample interaction. The shift in the phase angle of the cantilever vibration implies energy dissipation in the tip–sample ensemble, so it depends upon the specific mechanical properties of the sample underlying the tip. For the measurements, cantilevers with elastic constants of 5 N/m and 42 N/m were used. Antimony-doped silicon probes (TAP150A, TESPA-V2, Bruker) with resonance

frequencies 150 kHz and 320 kHz, respectively, and nominal tip radius of curvature 10 nm were used. Phase images were acquired simultaneously with the topography.

## 4. Results and Discussion

### 4.1. Rheology

The below rheological plots illustrate consistent viscoelastic transition temperature (TR) trends among the binders (see Figures 3 and 4). It is worth noting that at low temperatures, PB shows a higher phase angle value when compared with aged samples; this indicates that PB has lower rigidity. The TR for each of the studied aged binders increased with ageing, which implies an increase in hardness. The effect of ageing on pristine bitumen shifted the TR depending on the duration of ageing; for instance, the TR values of the most-aged binders, BRTFOT-225 and BPAV, increased by approximately 25 °C in comparison with the neat bitumen. The phase angle, tan $\delta$, increased with temperature, signalling a reduction in material consistency, which means that the prevalent liquid-like character is inclined with temperature [41]. In order to have a clearer image of the bitumen TR range and the structural modifications, temperature-sweep experiments were plotted in terms of the elastic modulus $G'$. This experiment was conducted at a frequency of 1 Hz and at a constant heating rate (1 °C/min). The binder's elastic modulus was observed continuously during a temperature ramp.

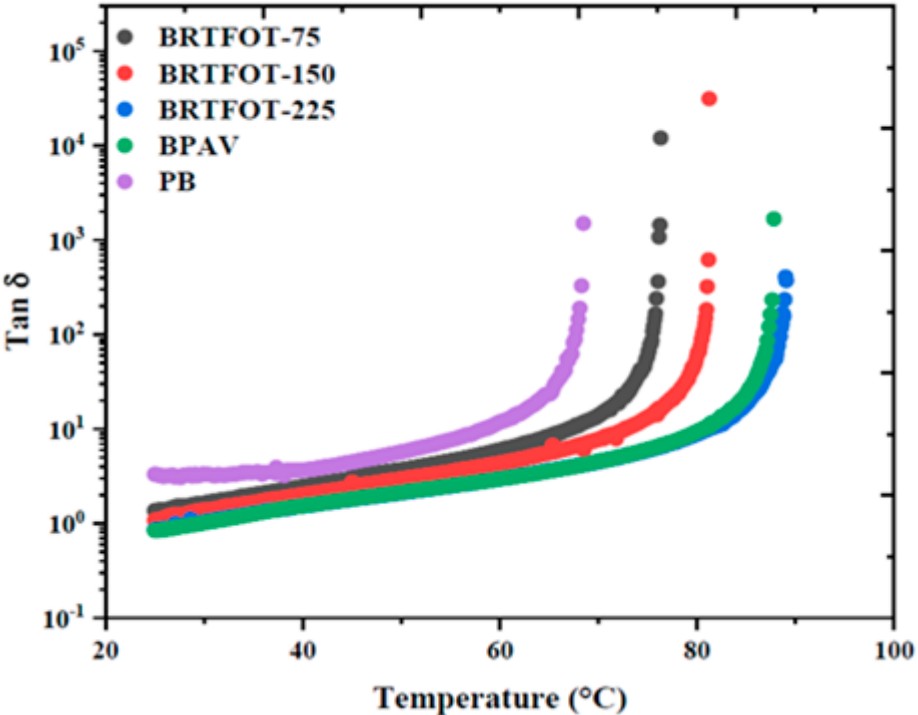

**Figure 3.** Temperature dependence (in the range 25 to 100 °C) of the loss tangent for pristine (PB) and aged bitumen samples (BRTFOT-75; BRTFOT-15; BRTFOT-225 and BPAV).

Figure 4 depicts the elastic modulus $G'(\omega)$ against temperature for all studied samples, in the temperature range 25 to 100 °C. In the temperature range 25–45 °C, the BRTFOT-75 and BRTFOT-150 binders had overlapped $G'(\omega)$ values, meaning their rigidity character is similar. At elevated temperatures, however, the aged binders showed higher $G'(\omega)$ values depending on their degree of ageing. For instance, PB had the lowest $G'(\omega)$ value in correlation with its tan $\delta$ value. After 60 °C for virgin bitumen and after 70 °C for aged binders, nonlinearity tended to appear in the trends, indicating the starting point of the TR region. When the elastic modulus is no longer detected, the transition

process from viscoelastic to a liquid regime can be considered complete, and this proves that the liquid-like behavior increases with temperature.

Interestingly, the viscoelastic responses of the PAV and RTFOT binders show similar rheological behavior. This analysis should prompt thinking that the two kinds of ageing processes (RTFOT 225 and PAV) could have similar effects on the structure and self-assembling properties of the binder.

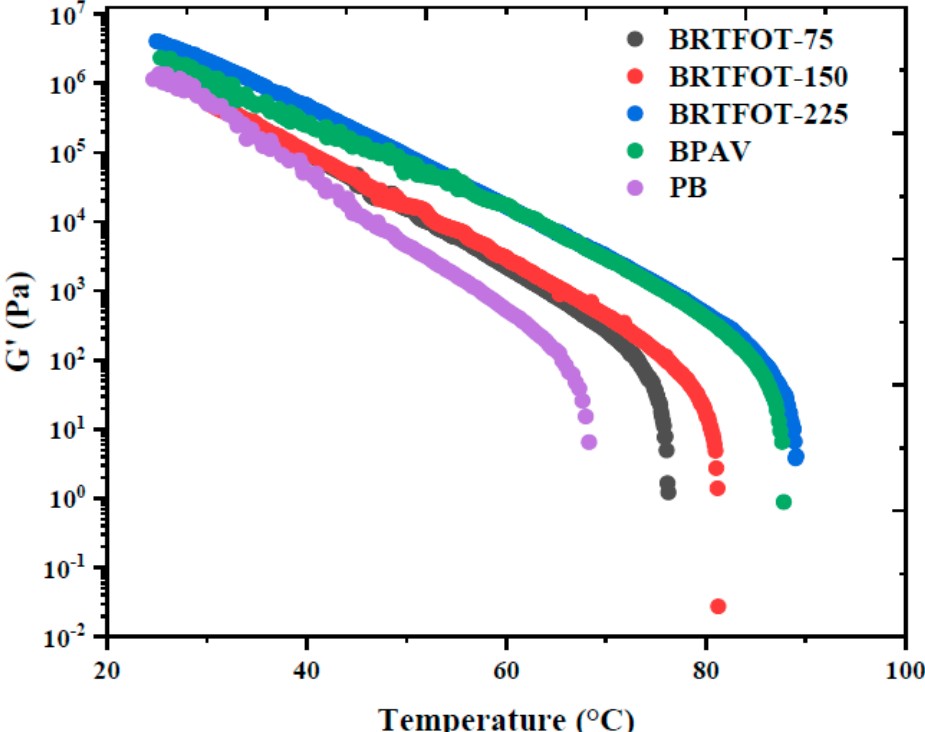

**Figure 4.** Temperature dependence (in the range 25 to 100 °C) of the elastic modulus for pristine and aged bitumen samples.

### 4.2. Optical Microscopy

The PB and BPAV samples were studied under optical microscopy since they were the softest and hardest samples among the investigated samples. In Figure 5, images of the PB and PAV asphaltenes are presented. The results imply that the BPAV binder is harder than the PB binder due to a higher concentration of asphaltenes.

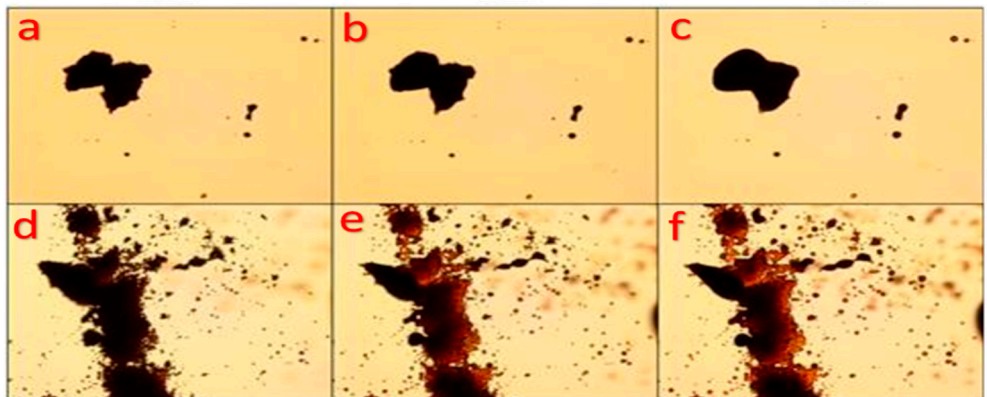

**Figure 5.** Optical microscopy (20×) images of PB (**a–c**) and PAV (**d–f**) asphaltenes at high temperature.

These images, obtained at 150 (a and d), 160 (b and e), and 165 °C (c and f), show the materials at different crystallization stages. The high-temperature stage of the experiment also shows the melting point of the crystallized material near 160 °C for the PB asphaltenes, while the material is partially melted at 165 °C for the BPAV sample, where the crystals are almost lost.

These novel results reveal two crucial points: First, the asphaltenes are solid at temperatures lower than 150 °C. Secondly, the asphaltene phase of virgin bitumen holds a different assembling structure to aged bitumen.

### 4.3. Fourier Transform NMR Self-Diffusion Coefficient Test

An advanced molecular investigation was conducted using NMR in order to analyze the chemical ageing processes and microstructural modification. As previously mentioned, Fourier transform (FT) NMR self-diffusion coefficient (SDC) allows insight into the bitumen microstructure by detecting the long-range mobility of the mixture constituents. The determination of motion over long distances, in comparison with ideal micelles, provides a sensitive probe for the state of aggregates [47,51]. It is crucial to note that the SDC values of asphaltene molecules cannot be detected owing to the short transverse relaxation times of their protons; so, hereafter, the measured SDC values are related to the maltene phase.

The observation of self-diffusion is based on the mobility of molecules. The motion of these molecules can be hindered due to the obstruction they face during their mobility in media. These typical data, observed where the motion occurs, can be considered as a fingerprint of the microstructural behavior.

In this study, the SDC data for each sample were investigated within the temperature range of 90–130 °C, increasing in increments of 10 °C at a time, as per Figure 6. The SDC data are related to the maltene part since at this temperature the asphaltenes are solid, meaning that their signals are not visible on the NMR spectrum after a spin echo sequence. According to this chart, the SDC values for binders decreased with ageing at each specific temperature. This effect is due to the stronger rigidity in aged bitumen which unavoidably involves stronger intermolecular connectivity when compared to the less dense network of the virgin bitumen, where the asphaltene domains are poorly connected to each other. As a result of this aggregation-based process/modification, a progressive shift from the viscoelastic toward the liquid regime dominating the highest temperatures occurs. In this picture, the resin molecules, due to their amphiphilic nature, will tend to reduce the associative interactions between the asphaltene particles by interposing themselves between the asphaltenes and the maltenes. This phenomenon might be like that found in the deactivation of hydrophobic cross-links in hydrophobically modified polymers by surfactants [43].

The interaction of the apolar part of the resin (its apolar moiety) with the maltene phase drags the latter to more-hindered dynamics typical of the stiffened asphaltene-dominated structure. Another mechanism concurrently present may be the formation of direct interactions between the surfactant polar headgroup and polar parts of asphaltene. In fact, it has been recently highlighted that, in addition to polar and apolar interactions, further specific interactions between surfactants themselves with consequent peculiar self-assembly processes [52,53] dictate the final overall aggregation pattern [54] and the (usually slowed-down) dynamics [55,56]. As an overall result of the two processes above described, maltene molecules in aged binders showed lower SDC values. What remains to be seen is the reproducibility of this gradual difference between virgin and aged binders at even higher temperatures in future experiments. It is worth noting that the BPAV bitumen's SDC value is a little bit higher than that of BRTFOT-225. This result could be due to the different dimensions of the asphaltenes or different structural network morphology. The BRTFOT-225 asphaltenes are expected to be bigger and to provide a strong obstruction to the maltene molecules' mobility. This difference in behavior is also visible in both ageing techniques (RTFOT and PAV). This is an interesting finding, and it could be hypothesized that the FT-NMR-SDC technique can be used to investigate the effects induced by the ageing phenomenon in bitumen. In order to confirm this hypothesis, AFM morphological images

were employed. By raising the temperature, we also observed upward trends in SDC values for all binders; this can be attributed to the incremental liquefying (melting) of the solid constituents, which inclines the motion of molecular protons. Although there was a steady rise in SDC measurements by a degree for each 10 °C increment, they increased dramatically at 130 °C, reaching twice the values of the respective binders at 120 °C. The results mentioned above are the most relevant findings and perhaps the most significant, confirming that the FT-NMR-SDC technique can be utilized as a fingerprint for the characterization of microstructural behaviors of bitumen.

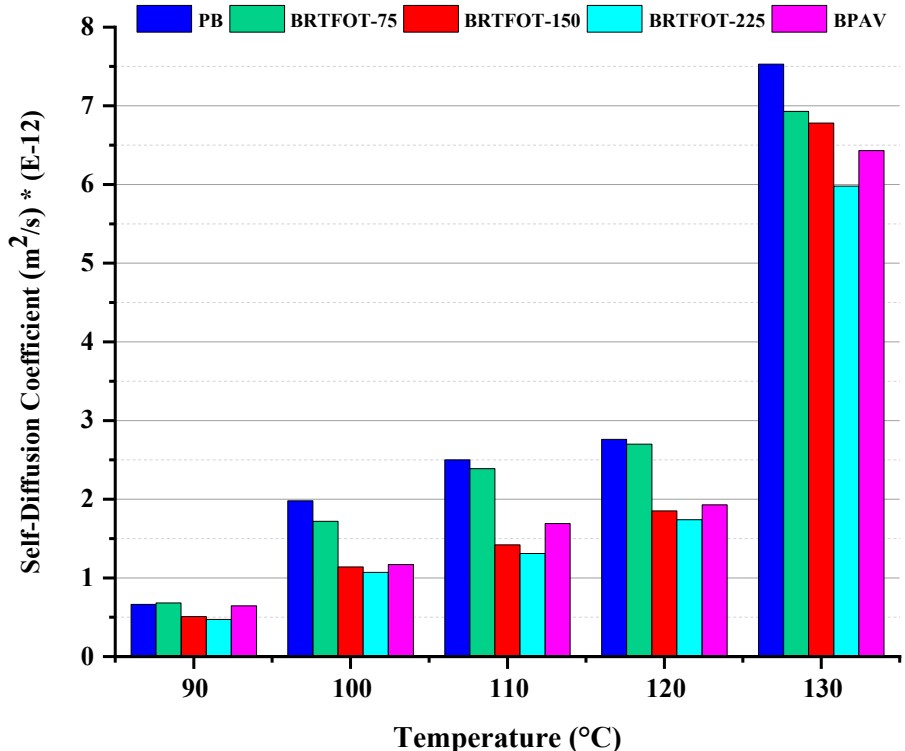

**Figure 6.** NMR self-diffusion coefficient characterization in the temperature range 90–130 °C for the different samples (PB, BRTFOT-75, BRTFOT-150, BRTFOT-225, and BPAV).

### 4.4. Atomic Force Microscopy (AFM)

AFM has been found to be a useful technique in bitumen microscopic analysis. In this work, we aimed to analyze the AFM images regardless of the correlations between the bitumen structures at different ageing processes in AFM images.

AFM characterization was performed in tapping mode at room temperature in air on a Multimode 8. The AFM system equipped with a Nanoscope V controller (Bruker) provided simultaneous topography and phase imaging of the sample. The measurements of bitumen were performed using probes with a conical tip of nominal end radius 10 nm and a resonance frequency of 150 kHz.

Phase images can show the sample's viscoelastic properties and are useful for bitumen microscopic analysis because materials with different viscoelasticity are clearly distinguishable: soft domains appear dark, while the hard ones appear bright.

The AFM measurements in Figure 7 show that with increasing exposure time (RTFOT), the morphological structure of asphaltenes changes. More specifically, in the aged samples, we note the arising of progressively more oscillations in the AFM signal within each single cluster, a feature which is known as "bee structure" [57] and which denotes structuring of asphaltene clusters at the micrometer-length scale [50]. Moreover, the asphaltene domain size increased with ageing time. For instance, at BRTFOT-225, the domains were about 10 microns in length and 5 microns in width;

the AFM image of PAV bitumen instead revealed asphaltene domains smaller than those observed in BRTFOT-225, homogeneously dispersed on the sample surface.

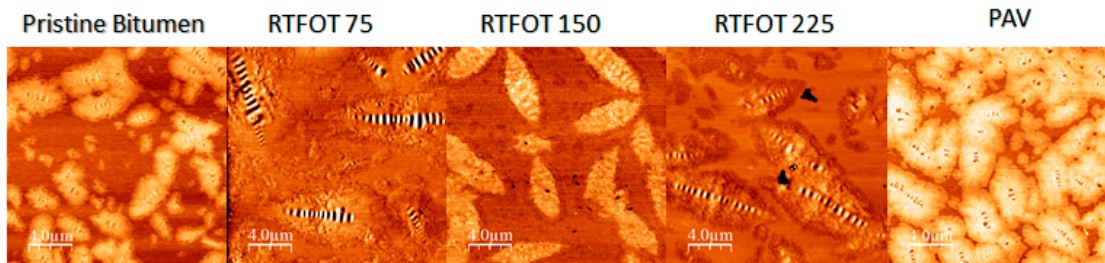

**Figure 7.** AFM phase images of PB, BRTFOT-75, BRTFOT-150, BRTFOT-225, and BPAV.

It must be pointed out that asphaltene clusters as they are seen by optical microscopy (length scales of micrometers) are the overall, long-range result of asphaltene molecules' hierarchical aggregation occurring at different length scales [50], for which different kinds of intermolecular interactions, from strong and short-range to weak and long-range, are involved. Among these, however, polar–polar interactions are expected to play the major role, since amphiphilic molecules can effectively bind, through their polar moiety, the polar groups of the dispersed molecules or the interfacial polar groups if the molecules are dispersed as clusters, as in the case of asphaltenes in the bitumen. So, a change of the cluster structure can be expected if the polar interactions are strong enough to trigger competition between asphaltene–asphaltene interactions and asphaltene–additive interactions, competition which has been found to be typical at the nanoscale in complex systems [58]. In light of this observation, atomic force microscopy can be seen additionally as a technique to indirectly probe the interactions taking place at shorter length scales, allowing us to explain, in the present work, the trends observed by FT-NMR-SDC. AFM images show the structural morphology of the bitumen, and they allow us to understand where maltene molecular mobility occurs. Considering the RTFOT ageing, the decrease in self-diffusion coefficients at all temperatures is apparently due to an increase in the size of asphaltenes, while the BPAV image clearly shows a different structural morphology from that in the image of BRTFOT-225.

## 5. Conclusions

In summary, ageing produces fundamental modifications in the colloidal structure of bitumen. Ageing also causes oxidation of bitumens and, consequently, increases the content of large molecules and the bitumen's molecular weight; these changes increase with ageing time. The two conventional ageing processes (RTFOT and PAV) modified the colloidal system of the bitumen, and both showed a certain level of asphaltene content. The RTFOT ageing, for different exposure times, changed the asphaltene structure in a consistent way, while the PAV ageing created a new colloid structural network. From the tangent loss and elastic modulus results, the rheological properties of the aged binders were found to be dependent on bitumen oxidation and changes in microstructures. The impact of these varies with the amount of ageing.

From the optical microscopy results, we concluded that the microstructural assembly of asphaltenes in unaged and aged bitumen is different, which is consistent with our FT-NMR-SDC observations. The core of this investigation was focused on using FT-NMR-SDC to track changes in the chemical functionalities of an unmodified binder and four types of aged binders manufactured with the same bitumen subjected to RTFOT/PAV ageing conditions. Most interestingly, this technique showed strong ability to monitor the ageing processes and to highlight the structural differences induced during ageing. The AFM results indirectly confirmed what was obtained via the FT-NMR-SDC technique by showing that the size of asphaltenes in aged bitumens is increased when compared with the size of asphaltenes in unaged bitumen.

The latter results provide new information in addition to that already present in the scientific literature regarding the subject, providing a more complete picture of the ageing process, in particular, on how the ageing techniques used in the laboratory affect the chemical structure of bitumen. This new information can be used in future research for the creation of new rejuvenating additives that are even higher performance than those currently on the market.

**Author Contributions:** Conceptualization, M.P. and C.O.R.; Writing-Original draft preparation P.C. and D.S.; Writing-review and Editing M.P.; Formal analysis V.L. and M.P.D.S.; Investigation P.C. and M.P.D.S.; Supervision C.O.R. All authors have read and agreed to the published version of the manuscript.

**Funding:** This research received no external founding.

**Conflicts of Interest:** The authors declare no conflicts of interest.

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
