# Peer review of "NMR Diffusiometry Spectroscopy, a Novel Technique for Monitoring the Micro-Modifications in Bitumen Ageing"

_applsci, doi:10.3390/app10165409_

Round 1

Reviewer 1 Report

The paper concerns the asphalt bitumen aging evaluated in laboratory through different methods. The main objective of the paper is to present a novel technique – NMR diffusiometry spectroscopy - of bitumen aging in comparison with another current methods. This is an important topic for research.

The paper is not original but it presents some novelty regarding the main purpose and the new method for aging evaluation.

Some minor revisions are recommended:

1 - The standard tests used in the rheological characterization are not presented. Methodology should be more descriptive. Please include more information regrading the materials properties.

2 - Any reference of Figure 2 was found in the text.

3 - Caption of Figure 5 refers the image g) but it does not exist. Similarly, image d is not referred.

4 - The scale is not visible in the AFM phase image of PB in Figure 7.

5 - Conclusions should be more detailed.

Author Response

Point-by-point response to the Reviewers’ comments

Ms. Ref. No.: applsci-878240

Title: NMR Diffusiometry Spectroscopy, a Novel Technique for Monitoring the Micro-modifications in Bitumen Ageing Corresponding Author:  Cesare Oliviero Rossi

Authors: Paolino Caputo, Dlshad Shaikhah, Michele Porto, Valeria Loise , Maria Penelope De Santo, cesare Oliviero Rossi.

Dear Editor

Applied Sciences

Thank you for sending the reports of Reviewers 1 and 2. Please find our point-by-point replies to their observations, which are ordered in the way they posed the comments. Changes to the original manuscript are highlighted in yellow in the revised form. The authors are grateful to the Reviewers for their useful suggestions that helped to improve the present manuscript.

Reviewer #1

The paper concerns the asphalt bitumen aging evaluated in laboratory through different methods. The main objective of the paper is to present a novel technique – NMR diffusiometry spectroscopy - of bitumen aging in comparison with another current methods. This is an important topic for research.

The paper is not original but it presents some novelty regarding the main purpose and the new method for aging evaluation.

Some minor revisions are recommended:

1 - The standard tests used in the rheological characterization are not presented. Methodology should be more descriptive. Please include more information regrading the materials properties.

According to the Referee’s observation, we have extended the rheological description evidencing as the rheological measurements can be used normally as standard methods. We added more information about the materials.

2 - Any reference of Figure 2 was found in the text.

We agree with the referee and we added the reference of fig 2 in the text

3 - Caption of Figure 5 refers the image g) but it does not exist. Similarly, image d is not referred.

We corrected the text

4 - The scale is not visible in the AFM phase image of PB in Figure 7.

We added the scale in the PB image

5 - Conclusions should be more detailed.

We extended the conclusion paragraph

Reviewer 2 Report

The manuscript is globally well written. The authors have studied the characteristics of artificially aged bitumen. Two techniques were used to simulate both the short- and long -term bitumen ageing processes. A technique based on the NMR spectroscopy was used for the determination of the the self-diffusion coefficients for different aged samples. Results presented by the authors as the most interesting in their opinion were compared to the AFM ones. The work of the authors may help improving the permformace of rejuvenating additives which are actually used.

There authors may/should pay attention to the following points:

  1. There is a nuance between the two spelling "aging" and "ageing", check if the spellings are correctly used.
  2. Introduction, line 4. Replace "consits" by "consisting of" to avoid the use of two verbs in the same sentence.
  3. Caption of fig.1. Explain the abbreviation SARA.
  4. Subsection 3.1. What is phi and its dimensionb, unit?
  5. Remove "is the measurement of" in front of the two parts of G(omega).
  6. Subsection 3.2. Explain the abbreviation DMLP. Remove e"which" in the first line.
  7. Fig. 2. Indicate the x- and y-labels, units
  8. Text after fig 2. rf-->RF
  9. Around eq 2. It is not clear the differences between T_1, T_2, tau_1, tau_2 and tau_m. indicate what do I and I_0 stand for.
  10. In fig. 4, the G' values are in units of Pa. You should indicate that when you first present the real and imaginary parts of G(omega).
  11. Subsection 4.3. The sentence "This effect cannot but being....each other" is not clear. To rephrase.
  12. What do you mean by molecule protons?

Author Response

Reviewer #2

The manuscript is globally well written. The authors have studied the characteristics of artificially aged bitumen. Two techniques were used to simulate both the short- and long -term bitumen ageing processes. A technique based on the NMR spectroscopy was used for the determination of the the self-diffusion coefficients for different aged samples. Results presented by the authors as the most interesting in their opinion were compared to the AFM ones. The work of the authors may help improving the permformace of rejuvenating additives which are actually used.

There authors may/should pay attention to the following points:

  1. There is a nuance between the two spelling "aging" and "ageing", check if the spellings are correctly used.

We thank the referee and we used ageing term trough out the text of the paper

  1. Introduction, line 4. Replace "consits" by "consisting of" to avoid the use of two verbs in the same sentence.

We did it

  1. Caption of fig.1. Explain the abbreviation SARA.

We did it

  1. Subsection 3.1. What is phi and its dimensionb, unit?

The referee is right, we specified the meaning and the units.

  1. Remove "is the measurement of" in front of the two parts of G(omega).

We substituted it with represents

  1. Subsection 3.2. Explain the abbreviation DMLP. Remove e"which" in the first line.

We removed “which”

  1. Fig. 2. Indicate the x- and y-labels, units

We did it

  1. Text after fig 2. rf-->RF

We did it

  1. Around eq 2. It is not clear the differences between T_1, T_2, tau_1, tau_2 and tau_m. indicate what do I and I_0 stand for.

We specified all terms involved in the equations 2

In fig. 4, the G' values are in units of Pa. You should indicate that when you first present the real and imaginary parts of G(omega).

We specified it before the figure 4

  1. Subsection 4.3. The sentence "This effect cannot but being....each other" is not clear. To rephrase.

We did it

  1. What do you mean by molecule protons?

We corrected with molecular protons.
